# A Comprehensive Analysis of Factors Associated with Intimate Partner Femicide: A Systematic Review

**DOI:** 10.3390/ijerph19127336

**Published:** 2022-06-15

**Authors:** Esperanza Garcia-Vergara, Nerea Almeda, Blanca Martín Ríos, David Becerra-Alonso, Francisco Fernández-Navarro

**Affiliations:** 1Departament of Quantitative Methods, Universidad Loyola Andalucia, Avenida de las Universidades s/n, 41704 Seville, Spain; dbecerra@uloyola.es (D.B.-A.); fafernandez@uloyola.es (F.F.-N.); 2Departament of Psychology, Universidad Loyola Andalucia, Avenida de las Universidades s/n, 41704 Seville, Spain; nmalmeda@uloyola.es; 3Departament of Legal and Political Sciences, Universidad Loyola Andalucia, Avenida de las Universidades s/n, 41704 Seville, Spain; blancamartin@uloyola.es

**Keywords:** intimate partner homicide, femicide, violence against women, factors, systematic review

## Abstract

There has been a growing concern about violence against women by intimate partners due to its incidence and severity. This type of violence is a severe problem that has taken the lives of thousands of women worldwide and is expected to continue in the future. A limited amount of research exclusively considers factors related only to these women’s deaths. Most focus on deaths of both men and women in an intimate partnership and do not provide precise results on the phenomenon under study. The necessity for an actual synthesis of factors linked solely to women’s deaths in heterosexual relationships is key to a comprehensive knowledge of that case. This could assist in identifying high-risk cases by professionals involving an interdisciplinary approach. The study’s objective is to systematically review the factors associated with these deaths. Twenty-four studies found inclusion criteria extracted from seven databases (Dialnet, Web of Science, Pubmed, Criminal Justice, Psychology and Behavioral Science Collection, Academic Search Ultimate, and APA Psyarticles). The review was carried out under the PRISMA guidelines’ standards. The studies’ quality assessment complies with the MMAT guidelines. Findings revealed that there are specific factors of the aggressor, victim, partner’s relationship, and environment associated with women’s deaths. The results have implications for predicting and preventing women’s deaths, providing scientific knowledge applied to develop public action programs, guidelines, and reforms.

## 1. Introduction

Intimate partner violence is a serious social and public health problem affecting millions of females worldwide [1,2,3]. Historically, it has been conceived as a private family matter, but there has been growing concern about its high incidence and severity [4,5]. Violence against women by intimate partners refers to physical, sexual, and psychological assault women suffer from their current or former partners [6,7]. In the most extreme cases, it results in women’s deaths. Global estimates of this victimization indicate that approximately 38.6 percent of all homicides against women are committed by their intimate partners [8]. This amounts to over 30,000 females murdered annually [9].

The severity of violence has attracted the attention of numerous researchers. Many have studied factors associated with these deaths to obtain comprehensiveknowledge about the phenomenon to predict and prevent it [10,11]. Nevertheless, the deaths of women at the hands of their intimate partners are still being recorded and cannot yet be averted [12,13]. For this article, we define Intimate Partner Femicide (IPF) as those deaths suffered by women and inflicted by their present or former intimate partners in heterosexual relationships [14,15,16,17]. The studies of factors associated with IPF are not enough [18,19]. Some cases of IPF do not match with specific associated factors found in the current scientific literature. Some of them have been assessed and reported as having a low risk of serious violence [20,21,22]. In this regard, several cases of IPF present different factors from those currently known and, therefore, from those included in risk assessment instruments [18,19]. Adequate identification of this violence allows effectively dealing with it [23]. The research evidence has recognized a continuing need to study risk factors for IPF [7].

Multiple researchers have integrated the results of different studies about risk factors of intimate partner homicide in systematic reviews and meta-analyses [24,25,26]. However, these studies do not distinguish the sex of the victim and aggressor, including integrally as the study population the homicides from men to women, women to men, men to men, and women to women. This scope is too broad to obtain comprehensive knowledge of IPF [27,28]. These deaths go beyond the specific context of men killing women in intimate partner relationships due to the patriarchy and machismo history that has placed men in a position of superiority and dominance over women [29]. According to the feminist approach, the mentioned gender inequality is the basis of violence against women and the explanation for the significant difference in the death numbers of women and men in intimate partner relationships [30,31,32]. For the mentioned reasons, specific synthesis factors in this population are fundamental.

The review of Contreras (2014) [18] and the meta-analysis of Spencer and Stith (2020) [7] focused solely and exclusively on studies of risk factors for IPF and could provide more concise information on the phenomenon. The most relevant findings of these studies reveal that threats to harm, forced sex, substance abuse, controlling behaviors, strangulation, maltreatment during pregnancy, and a history of mental health problems are factors associated with IPF. However, the review results are not updated because, since their completion, other scientific studies on this topic have also been published [33,34]. The findings of the meta-analysis could be extended to a more significant number of factors, including qualitative and mixed studies, and not only quantitative. Additional studies on factors associated with IPF are also relevant to identifying elements related to the phenomenon that are not considered risk factors so far. This could guide future studies on risk factors. The meta-analysis is focused on aggressor and victim factors, being necessary to expand on the relationship between them and environment-related factors. It is considered relevant to analyze them as independent categories of analysis since personal characteristics influence behavior, dynamics with other people, and the environment where they are [30,35,36,37].

Studies on IPF have commonly focused on the offender and the victim to understand the criminal behavior, attributing less importance to the place where the crime happens [38,39]. On cases other than IPF, several studies have found a significant relationship between environmental factors and criminal acts, evidencing that timing and location are critical in the perpetration of the crime [40,41]. Therefore, it is also important to analyze the context in which IPF occurs.

Going back to crimes in general, having an intimate relationship influences the chances of crime perpetration. On the one hand, an intimate relationship protects the committing of crimes with factors such as partner support. On the other hand, stress coming from conflict between partners can potentiate it [42,43,44].

A new systematic review that addresses these aspects and integrates the current scientific studies of factors associated with IPF is necessary, hence the present work. It aims at synthesizing factors associated with IPF identified by the scientific evidence. The research questions are: (1) What are the aggressor-associated factors? (2) What are the victim-associated factors? (3) What are the partner’s relationship-associated factors? (4) What are the environmental-associated factors?

## 2. Methodology

The systematic review was performed under the Preferred Reporting Items for Systematic Reviews and Meta-Analysis (PRISMA) guidelines [45]. The PRISMA guidelines include a 27-item checklist with details about information required to ensure a quality scientific review. In addition, it includes a four-phase flow diagram that synthesizes the identification, screening, eligibility, and inclusion to exhibit the systematic review process [46]. Even though the guide was initially used in the health framework [45], it has been adapted and applied to other areas of research such as violence [47,48,49]. A meta-analysis has not been performed because the statistical outcomes in the publications included in the current study are not sufficiently homogeneous for comparison.

### 2.1. Search Strategy

The search strategy was conducted on 14 October 2021, by EGV, in the following databases: Dialnet, Web of Science (WOS), Pubmed, Criminal Justice, Psychology and Behavioral Science Collection, Academic Search Ultimate, and APA Psycarticles. These quality databases are in line with the theme of the study. The search terms included in the mentioned databases were made of three sets of keywords combined with different Boolean operators: (“gender violence” OR “gender-based violence” OR “intimate partner violence” OR “domestic violence” OR “intimate partner aggression” OR “violence against women”) AND (“homicide” OR “mortality” OR “kill” OR “intimate partner homicide”) AND (“factors” OR “characteristics” OR “causes”). These terms were identified in a thesaurus and in different studies on the same theme [7,24]. The search was limited by the title and/or abstract-mentioned terms. No limitations of place or time were applied.

### 2.2. Eligibility Criteria

Studies were included in the systematic review if they (1) identify personal characteristics of aggressors, victims, and/or relationships of IPF, (2) detect environmental factors associated with IPF, (3) are empirical articles, (4) are in Spanish or English language, and (5) are accessible in full text.

Studies were excluded if they (1) identify only factors associated with non-lethal intimate partner violence, (2) develop typologies of offenders of intimate partner homicide, (3) analyze risk assessment instruments of intimate partner homicide and not factors associated with this phenomenon, (4) examine case studies, and (5) incorporate homicides in same-sex couples.

The mentioned inclusion and exclusion criteria were applied, after removing duplicate studies, from all those identified in the database search. In particular, studies were initially screened by title and abstract, excluding those matching the exclusion criteria. The remaining studies were full-text read, and those that did not meet the inclusion criteria were excluded. Finally, the studies that met the inclusion criteria mentioned above were included.

### 2.3. Quality Assessment

To evaluate the quality of the scientific studies included in the systematic review, the Mixed Methods Appraisal Tool (MMAT) guidelines [50,51] were used. It includes a checklist with different items to assess the quality of quantitative, qualitative, and mixed studies in a systematic review. It is made of 2 general items that assess the adequacy of the research questions in all the categories of methods studied. It also has seven specific items that evaluate the appropriateness of data collection and the analysis of each category. The mentioned guideline is a unique, efficient appraisal tool focused on the methodology criteria of different study designs simultaneously [50]. For this reason and given that this quality assessment tool has been applied to systematic reviews related to the intimate partner violence topic [52,53,54], it has been selected for the quality assessment. Two researchers assessed the quality of each study, and a third one resolved discrepancies.

## 3. Results

### 3.1. Search Results

The search strategy yielded 1186 publications across all databases and two from other sources (reading the reference lists of the publications obtained on the databases). There were 558 removed as duplicates, and 630 remained. The title and abstracts of these were read, and 536 were removed according to the inclusion and exclusion criteria. The remaining 94 were full-text read, and 70 were removed according to the same eligibility criteria. Finally, 24 were included in the systematic review (see Figure 1).

### 3.2. Study Characteristics

The characteristics of the included studies are summarized in Table 1. It contains information about the country, year, sample, data sources, and main findings of factors related to IPF. The following paragraphs summarize this information.

#### 3.2.1. Country and Year

The first study was published in 1999 in the United States [55], and it has continued in subsequent years. A total of eight studies were located in the U.S., in each of the following years: 1999 [55], 2003 [13], 2004 [56], 2016 [11], 2019 [57], 2020 [34], 2021 [58], and 2022 [59]. Four studies were conducted in the United Kingdom in 2011 [1], 2016 [60], 2017 [61], and 2020 [2]. Three studies were conducted in Portugal in 2016 [33], 2019 [6], and 2021 [10], as well as three in Spain in 2005 [62] and 2019 [63,64]. Two studies were released in Norway in 2013 [65] and 2019 [66]. Only one study was performed in the following countries: Sweden in 2004 [67], Finland in 2012 [68], Brazil in 2021 [69], and Canada in 2021 [70].

#### 3.2.2. Sample

The study samples were divided into two groups. First is female victims of violence by current or former male partners in heterosexual relationships [2,11,13,34,55,56,57,58,59,65,69]. These male perpetrators constitute the second group [1,6,10,33,60,62,63,64,67,68]. The size of the sample varied among the studies between 8 and 151,826 participants for the first group [11,69] and from 104 to 854 for the second group [1,67]. There are some studies in which both groups form the sample [61,66,70], and their sizes range from 93 to 207 participants [61,70].

#### 3.2.3. Data Sources

Nine studies collected data by interviews and questionnaires, inventories, or surveys [6,11,13,34,56,58,59,62,65]. The most common instruments used were the Danger Assessment, the Brief Symptoms Inventory, the Marital Violence Inventory, the Buss-Perry Aggression Questionnaire, the Hare Psychopathy Checklist, the Conflict Tactics Scale-2, the Interpersonal Reactivity Index, the Violence Risk Appraisal Guide, and the Spousal Abuse Risk Assessment [71,72,73,74,75,76,77,78,79]. Nine additional studies were based on official databases from police, forensic, judicial, health, social, and educational services. Some studies specified the database such as the Finnish Homicide Monitoring System Database, the USA National Violent Death Reporting System, and the Ontario Domestic Violence Death Review Committee [2,57,63,64,66,67,68,69,70]. Six studies combined interviews, questionnaires, and official data [1,10,33,55,60,61].

### 3.3. Main Findings

There are specific factors of the aggressor, victim, partner’s relationship, and environment associated with IPF. The factors reported by each study are detailed in Table 1.

#### 3.3.1. Aggressor-Related Factors

There are socio-demographic characteristics of aggressors associated with IPF. These are age, education level, employment situation, socio-economic status, and ethnicity. The age difference between man and woman matters in IPF, it being considerably common for aggressors to be older than the victims [6,61]. Elementary education and low-medium socio-economic status are also factors associated with perpetrating this crime [62]. The link is even greater if the men in question are unemployed and receive neither unemployment benefits nor a pension [33,62]. Furthermore, being a stay-at-home spouse is also connected with IPF [34,61,68].

Immigration could be associated with IPF because; in some cases, this condition entails several risk factors such as being a member of an ethnic minority group, unemployment, lack of economic resources, low socio-economic status, low education, and excessive stress [33,34,61,62,64,67]. These factors are especially predominant in aggressors who have suffered pre-migration trauma [70]. The immigration factor becomes stronger in the association of death when the victim is also an immigrant and comes from the same ethnic background [67]. The connection of the immigrant factor with IPF is more common in non-recent immigrants than recent immigrants, so the immigration factor has not always had the same degree of association with IPF [70].

**Table 1 ijerph-19-07336-t001:** Summary of the factors associated with IPF according to the scientific studies.

Ref.	Country	Sample Number	Sample Characteristics	Methodology	Instrument Source	Factors Associated with IPF
[56]	USA	53 women	Women recruited from two prenatal care clinics in North Carolina (USA) who suffered physical violence by their male partner during prenatal routine care	Prospective study	Interviews using the Danger Assessment Instrument [71]	Drug abuse, jealousy, violent acts, controlling acts, death threats, separation during pregnancy months, violence the year before pregnancy, and controlled acts
[1]	U.K.	104 men	Men from Britain prisons convicted of murdering a marital, ex-marital, girlfriend, ex-girlfriend, or serious dating partner	Retrospective study	Interviews and official police, forensic, judicial, health, social, and educational data	History of violence on previous intimate partners, relationship problems, authority and control needs, strong cognitions bias about subordinate position of women to men and its normalization, possessiveness, jealousy, fear of abandonment, cognitions that justify the violence and minimize its severity and denial of the responsibility, lack of empathy and remorse, history of serious violence, early, persistent, and severe violence, separation, and couple’s relation characterized by conflicts, possessiveness, and controlling acts
[67]	Sweden	854 men	The sample was collected by 164 male perpetrators of spousal homicide and 690 other homicides committed from 1990-1999, recruited from the Sweden Police Register	Retrospective study	Official police and forensic data	Substance abuse, immigration, criminal records, psychiatric diagnose, separation, threats, and home
[33]	Portugal	187 men	Men convicted of violence against women recruited from different institutions of Portugal (50 men committed severe violence and 137 less severe violence)	Retrospective study	Official case files, interviews, and questionnaires, The Brief Symptoms Inventory [72], The Marital Violence inventory [73], the Buss–Perry Aggression Questionnaire [74], and the Hare Psychopathy Checklist-Revised [75]	Low–medium socio-economic status, use of guns, separation, previous intimate partner violence, threats with guns, injuries that need medical assistance, and persecuting acts
[68]	Finland	836 men	Men who kill women, women who kill men, men who kill men, women who kill women, and men and women who kill family members	Retrospective study	Official from Finnish Homicide Monitoring System database	Unemployment or pension, alcohol and/or drug abuse, knowledge of becoming violent when he is intoxicated, criminal records and judicial convictions, and previous intimate partner violence
[13]	USA	30 women	USA female survivors of attempted homicide by an intimate partner for the years 1994-2000	Retrospective study	Interviews	Jealousy, controlling acts, injuries, social isolation, desires for separation or divorce, history of violence, escalating frequency and severity of violence, death and injury threats with guns, and stalking
[11]	USA	8 women	Women who have experienced attempted homicide by their partners	Retrospective study	Interviews	Using weapons, sexual violence, controlling acts, extreme jealousy, prior intimate partner violence, physical injuries, strangulations, and death threats with guns
[57]	USA	2613 women	Women killed by their intimate partners between 2005 and 2013, cases perpetrated in rural and urban areas	Retrospective study	Official data from the USA National Violent Death Reporting System	Using firearms, high opposition to the former woman, multiple wounds and injuries to the face, head, and neck, and rural area
[61]	U.K.	207 men and women	207 male and female offenders and victims’ cases of intimate partner homicide between 1998 and 2009	Retrospective study	Interviews and official police data	Criminal convictions, men older than the female partner, drug and alcohol abuse, unemployed, housewife/husband or retired, partnership over 3 and below 10 years, married couple, and stepchildren
[34]	USA	266 women	Female victims of intimate partner violence between 2009 and 2010	Prospective study	Victim interviews using The Conflict Tactics Scale-2 [76] and Danger Assessment [80]	Immigration, unemployment, and arrest records
[2]	U.K.	25 women	Female victims of intimate partner homicide between 2005 and 2020 selected using the Counting Dead Women database	Retrospective study	Media report and documentaries, official judicial data, and interviews	History of control patterns, criminal and arrest records, history of domestic abuse, progressive possessiveness and control, imaginations of the separation, cognitive justifications, perception of lost control of partnership, purchase weapons, attempts of isolation, compliance of coercive control demands, advertises and desires of separation or divorce, history of violence, escalation of frequency, severity, and variety of violence, stalking, sexual violence, extreme subordinate relationship, separation, threats, and ignorance of friends
[60]	U.K.	105 men	Murders of female intimate partners from prisons, which were divided in two groups: first, men with previous conviction and, second, men without it	Retrospective study	Interviews and official police, judicial, social, educational, and health data	Possessive, rationalizations and justifications for violence, family problems in childhood, behavioral and/or learning problems at school, physically abused in childhood, drug and alcohol abuse, history of criminal acts, sexual problems, lack of empathy, separation, cohabiting, serious relationship, ongoing disputes, history of violence, and sexual violence
[66]	Norway	177 men and women	Victims and aggressors with and without drug and/or alcohol abuse involved in intimate partner homicide from 1990 to 2012	Retrospective study	Official judicial data	Alcohol or drug abuse
[55]	USA	208 women	Killed and attempted victims of intimate partner homicide between 1994 and 1998	Retrospective study	Interviews, official judicial data, and stalking questionnaire [81]	Stalking, prior history of violence, and separation
[65]	Norway	157 women	Victims of intimate partner homicide from Norway from 1990–2012	Retrospective study	Interviews	Danger perceptions during the violence perpetration, severe and frequent violence, and threats of deaths
[63]	Spain	168 men	118 Spanish and 50 immigrant aggressors of intimate partner homicide between 2000 and 2011	Retrospective study	Official judicial data	Criminal records, stepchildren, partner discussions, separation, and home
[62]	Spain	162 men	Men serving a prison sentence for severe intimate partner violence or homicide	Retrospective study	Inventario de Pensamiento Distorsionados sobre la Mujer y el Uso de la Violencia [82], Interpersonal Reactivity Index (IRI) [77], Violence Risk Appraisal Guide (VRAG) [78], Sympton Checklist (SCL-90) [83], Psychopathy Checklist-Revised (PCL-R) [75], Trait-State Anger Inventory [84], and Barratt Impulsiveness Scale (BIS-10) [85]	Distorted ideas about women and about violence as an acceptable form to resolve problems, elementary education, low–medium socio-economic level, and separation or divorce
[64]	Spain	307 men	Men with a sentence for consummate or attempted intimate partner homicide between 2012 and 2015	Retrospective study	Official judicial data	Jealousy, excessive stress for denounces, knowledge or suspicionthat the female partner is with another man and economic problems, access to weapons, mental illness, separation, stalking, threats of death, and controlling acts
[6]	Portugal	172 men	137 aggressors of intimate partner violence and 35 of intimate parent homicide recruited from prison and community services	Retrospective study	Spousal Abuse Risk Assessment (SARA) [79]	Drug or alcohol abuse, suicidal ideation or intent, use of weapons, cognitive minimization or denial of violence, personality disorder, jealousy, men older than the women, threats of death, history of violence, escalation of violence, and marital status
[59]	USA	213 women	Victims of attempted intimate partner homicide	Retrospective study	New Jersey Assessment of Domestic Violence Risk and Impact (NJADVRI) [59]	Controlling acts, access to a gun, drug abuse, violent acts, jealousy, unsafe feelings, history of violence, increasing severity and frequency of violence, stalking, and threats of death
[10]	Portugal	245 men	Aggressors of intimate partner violence and intimate partner homicide recruited from prisons and community services	Retrospective study	Interviews, official judicial data, and the Brief Symptoms inventory (BSI) [72], the Marital Violence Inventory (IVC) [73], and the Hare Psychopathy Checklist-Revised (PCL-R) [75]	Use of weapons, criminal records, prior history of violence, separation or divorce, and no children
[69]	Brazil	151826 women	Victims of violence perpetrated by their intimate partners	Retrospective study	Official data from the Mortality Information System (SIM) and the Notifiable Diseases Information System (SINAN) of Brazil	Rural areas, history of violence in the partner relationship, physical, sexual, and psychological violence simultaneously, and use of weapons
[58]	USA	661 women	Victims survivors of intimate partner violence recruited from domestic violence shelters	Retrospective study	Survey instruments of Danger Assessment (DA) [86], Severity of Violence Against Women Sexual Violence Subscale [87], and specific items made based on the scientific literature	Reproductive coercion and pregnancy avoidance
[70]	Canada	93 women and men	Immigrant male aggressors and female victims of intimate partner homicide perpetrated between the years 2002 and 2016	Retrospective study	Individual case reports from the official database of Ontario Domestic Violence Death Review Committee (DVDRC)	Non-recent immigrants and pre-migration trauma

Regarding the biography of aggressors, the adult’s criminal record is associated with IPF [2,60,61,63,67,68]. Men who have been arrested and given a protection order or prison for an offense have more probabilities of perpetrating IPF [2,60,68]. However, arrest due to violence against women by itself is not significantly associated with a subsequent death [34]. The criminal records due to a violent offense against previous intimate partners or family members and/or violent conflicts with them, especially if violence was accompanied by controlling patterns from the offender, are more strongly associated with IPF [1,2,10].

Family problems during childhood are common amongst aggressors, and some of them have been physically abused by a family member during this period. This is another factor associated with the perpetration of IPF [60]. However, some aggressors have been victims or witnesses of family violence during their childhood and have not perpetrated violence against women in their adulthood [6]. However, several aggressors also had school problems in their infancy related to behavioral and learning problems, which are associated factors too [60]. These issues could also have an impact on the aggressor’s mental health, which is again a factor related to IPF [64]. Psychiatric diagnoses of affective disorder, psychotic disorder, and personality disorders are factors associated with IPF [6,67]. Suicidal ideations or attempts of aggressors, typical of affective disorders, are linked to IPF [67]. Substance abuse or dependence is another disorder associated with IPF [6,59,60,67,68], it being stronger when the aggressor’s consumption, despite being conscious of the drugs’ and alcohol’s effects, leads to more violent acts towards their partners, while they continue consuming drugs [68]. Furthermore, the connection of this factor with IPF is stronger when both aggressor and victim suffer substance abuse—including alcohol and/or drugs [67].

Mental disorders potentiate IPF when the aggressors have specific cognitions such as distorted beliefs about the subordinate position of women to them and the justification of violence to maintain it by cognitive neutralization techniques [1,62]. Men who use violence and blame the victim or environmental circumstances, minimizing or denying the damage caused to her by considering that it is necessary and believing that she deserves it, have more probability of killing their partners [2,6,60,62]. The rigid conceptions of men about authority, possessiveness, and control over women result in immense fear of abandonment of their partner through separation or divorce, perceiving lost control of her and their partnership, which is a factor associated with IPF [1,2,13].

Extreme jealousy of aggressors by the presence of the mentioned conceptions is a factor related to IPF as well [1,6,11,13,59,64]. The imaginary or real assumption that the victim is dating another man increases much more the strength of this association [64]. Additionally, the lack of empathy and remorse in the aggressor is also associated with IPF [1,60].

The dysfunctional cognitive schemas of aggressors are reflected in behavioral problems that predispose them to kill. In particular, the aggressors’ beliefs of the subordination of women to men lead to possessiveness and controlling acts over the victims [1,2,59,60]. The aggressors’ efforts to isolate the victim, the high opposition to their last partner, and the use of violence and weapons are forms of keeping their power and domain over the victims, which act as factors associated with IPF [2,11,57,59]. Reproductive coercion and pregnancy avoidance of the victim comprise another controlling act that leads to women with low reproductive decision-making power and is considered a factor associated with IPF [58].

The purchase of, access to, and use of weapons (especially guns) are factors associated with IPF and, even more, if the aggressor had used them in the past [2,6,10,33,57,59,64,69]. Mainly, the use of weapons by men in sexual offenses in which the victim is intimidated to obtain the desired sexual activity is strongly connected with IPF [11].

Sexual problems in men are linked to sexual crimes against their female partners, but also IPF [60]. The state of pregnancy of the victim combined with the aggressor’s previous behavior pattern characterized by drug abuse, violence, controlling and jealous acts, death threats to the victim, and violent acts against other people are factors strongly associated with IPF as well [56]. Violent behavior against people outside the couple’s family nucleus is not linked with IPF [6].

#### 3.3.2. Victim-Related Factors

As mentioned before, the age difference between men and women in relationships affects IPF. One characteristic of the victim that represents a factor associated with these deaths is being younger than the aggressor [6,61]. Nevertheless, the presence of this single factor is not enough to predict IPF, but it is in combination with other factors such as being an immigrant [67] and the consumption of alcohol and drugs by both the aggressor and the victim simultaneously [61,66].

The increase in the severity and frequency of violence against the victim, multiple injuries, an intense feeling of unsafety that gets her to consider that her partner would be capable of killing her, and close people perceiving the victim to be alarmed by the violent situation are factors strongly associated with IPF, particularly during the days following such an indication of alarm [13,59,65]. This association is greater when the injuries suffered by victims are on their face, head, or neck [57]. However, the absence of people’s perception of fear and alarm in the victim does not mean that the probability of homicide against the woman is lower, given that the victim could be isolated, which is a factor associated with IPF as well [13]. The isolation and employment status of victims are linked to IPF since those victims who do not have a job, are retired, or are housewives commonly find themselves isolated [61].

Women who are submissive to men’s demands lose their choice of freedom as the aggressors take control of their lives, posing a factor associated with IPF [2]. The loss of control even in the own partner’s relationship—for instance, the aggressor decides when they have sex, which leads to sexual offenses—is another factor connected with IPF [11]. However, the contradiction of the victim to the imposed submissive demands by separation desires and the communication to the aggressor to end the maltreatment are factors strongly linked with IPF, especially on the days after the communication [2,13].

#### 3.3.3. Relationship between Aggressor and Victim—Related Factors

The most important factor associated with IPF is a partnership characterized by repeated violence from aggressor to victim [1,2,11,33,59,65,68,69]. This factor increases the probability of death when its frequency and severity rise over time [6,13]. This aggravation is commonly associated with sexual violence. Thus, it is an indicator of the seriousness of their partner’s violence and could result in death [11,60]. However, recent studies add that this escalation is associated with various types of violence, such as physical and psychological violence. It is thus not only associated with sexual violence [2,13,59]. The presence of several types of violence—physical, sexual, and psychological—is simultaneously more associated with IPF than the occurrence of each of them separately [69]. Furthermore, early violence in the relationship is a predictor of later persistent and severe violence and homicide [1].

Of all typologies of violence, those that cause injuries that need medical assistance are strongly related to IPF [11,33], particularly injuries from violent acts of strangulation [11]. These violent offenses are not the only factors associated with IPF: injuries and death threats are also related [6,13,59,64,67], especially if they entail the use of weapons [11,13,33,65]. The escalation of violence with injuries and threats in combination is a potential indicator for a near-future homicide result [2].

Victims who are cohabiting with the aggressor at the same home suffer from more frequent violent acts, injuries, and threats, factors associated with IPF [60,61], mainly if the couple had been together for over three years and below ten years [61]. The victims who are married to the aggressor are more prone to be killed [6,61]. Nevertheless, girlfriends and boyfriends who have a serious partnership without marital status, but are living together, have a similar propensity [60]. Moreover, in such intimate partners, the presence of stepchildren who are not the biological offspring of the aggressor is another factor related to IPF [61,63].

Divorce or separation, followed by perceptions of abandonment by men who do not want to end the partnership, is another factor associated with IPF [1,10,33,62,64,67]. It is important to point out that the victim’s warning her intention to divorce or separate from the aggressor and the efforts made to leave the relationship are also associated factors, even if separation is not happening after all [2,60]. In addition, the association of divorce or separation with IPF is stronger during the pregnancy period of the victim, especially if the woman experiences a quick escalation of violence as soon as the aggressor knows about the pregnancy [56].

Stalking behavior is another factor associated with women’s deaths [13,59,64], it frequent being in aggressors who have divorced or separated recently [55]. The most common stalking behaviors that increase the probability of homicide are her being followed or spied on and him making repeated phone calls and waiting outside her house or workplace [33,55,59]. Besides, victims who have suffered physical abuse during the partnership are more likely to be stalked and, subsequently, murdered [2,55]. Consequently, the probability of a homicide is significantly higher when separation or divorce is followed by stalking behavior and prior violence [55].

A broken partnership is not the only problem in the relationship between the aggressor and the victim that is related to IPF: couple’s conflicts are also a factor associated with it [1,31,63]. Many of these conflicts are caused by the opposition of the victim to the extreme subordinate relationship of the aggressor, which is characterized by excessive power, control, and possessiveness over her [1,2,11,13,64]. The victim’s decisions and activities are controlled by the aggressor with coercive discourses in order to separate the victim from her family and friends; thus, the aggressor has more control over her [2,56,59]. These conflicts result in homicide within an ongoing conflict just as the aggressor perceives a loss of control over the victim and reacts impulsively [60]. However, arguments and disputes in a couple without violence are not associated with IPF [64].

#### 3.3.4. Environment-Related Factors

A significant association between location and IPF has been found [57,63,67]. Rural areas are associated with severe violence against women and IPF [57,69]. The geographical distance of the victims and their family members and close friends also matters in IPF perpetration. Victims who have been isolated by their aggressor, being moved to reside far from the homes of their family and friends, have a greater risk of homicide [13]. Furthermore, in cases where the aggressor’s friends are aware of the maltreatment and isolation that the victim suffers and do not take action to report it promote the result of homicide [2].

### 3.4. Quality Assessment Results

Quality is different across the quantitative, qualitative, and mixed method studies, which is presented in Table 2. Most quantitative method studies have appropriate samples and measures, clear research questions, and control of biases. A few studies do not present precise research questions, but they have clear research objectives and methodologies. With a few exceptions, all qualitative method studies have clear research questions, adequate and coherent data collection, qualitative approach, analysis, and interpretation of the findings. Those qualitative studies without clear research questions have clear research objectives and proper methodologies. The studies with mixed methods have clear research questions, adequate data collection, the use of mixed methods, and a proper integration and interpretation of quantitative and qualitative data.

## 4. Discussion

The objective of the systematic review was to “synthesize factors associated with Intimate Partner Femicide (IPF) identified by the scientific evidence to obtain comprehensive knowledge of it”. Generally, the results showed that there are factors of the aggressor, victim, partner’s relationship, and environment associated with these deaths. The findings validate and provide more detailed information on factors associated with IPF compared to previous systematic reviews [18]. For instance, concerning weapons, the findings reveal that everything related to them, including access, purchase, and common use, are factors associated with these deaths [2,10,33,57,59,64]. Another example is related to the rupture factor since separation or divorce is not only a factor associated with deaths due to the same association as the victim warning of her desire of rupturing the relationship to the aggressor even if it does not take place [2,18,60]. Moreover, the review contributes to the identification of a greater number of factors concerning previous systematic reviews [18]. It identifies new factors such as elementary education, low–medium socio-economic status, maltreatment in childhood, school problems, lack of empathy, remorse, stepchildren, and sexual problems.

The current systematic review also identifies additional factors associated that are not included in the meta-analysis of Spencer and Stith (2020) [7] such as immigration, arrest, protection order and prison, physical abuse during childhood, distorted beliefs, lack of empathy and remorse, victim’s unsafe feeling, victim’s submissive pattern, rural areas, and geographical distance. Furthermore, specifications are provided on factors already identified in this mentioned meta-analysis as age and length of intimate partner relationship. Young age is not relevant by itself due to aggressors being older than victims being associated with IPF [6,61]. The time of the relationship is general, and the current study specifies that serious partnerships with cohabitation between 3 and 10 years is associated with IPF [61].

There are theoretical explanations for the factors’ findings. Regarding attachment theory [88,89], some men with an anxious–insecure attachment pattern have killed their female partners by uncontrollable violent acts that were initially intended to avoid abandonment [12]. Men with this attachment believe that maintaining interpersonal relationships requires great effort and do everything possible not to lose those that they love [89]. This could explain the findings on break up, stalking, and jealousy factors. Additionally, some aggressors have an avoidant–insecure attachment that leads them to use violence to gain attention, power, and control [89,90]. The submissive pattern of victims could maintain the violence of aggressors with these attachments, leading to an escalation of violence that could end in death. This could explain why submissive and insecure feelings are considered factors associated with IPF.

Concerning psychopathological theories, some men with mental disorders have social and interpersonal problems, and they socialize through aggressive means, which could be, in exceptional cases, fatal [12]. Some of them are aggressors of IPF. This substantiates affective, psychotic, and personality disorders as factors associated with these deaths. It is important to note that people with mental disorders are not always homicide offenders. In addition, some men with mental illness suffer from substance abuse (drug and alcohol) to cope with their difficulties, which is another factor related to IPF [29,91].

Regarding sociofeminist theories, the violence against women is a manifestation of men who continue the patriarchal and machismo ideas [29]. They consider the use of violence necessary to dominate and control their partners [92]. This violence appears when females show opposition to the superiority of men within the family. In these cases, men try to keep their status even if it ends with the woman’s death [13,93]. This theory substantiates the factor of biased cognitions of violence and the subordination of women to men.

When it comes to social learning and intergenerational transmission of violence theories, violent behavior is learned by socialization and observation [94]. Thirty percent of people who have suffered or have witnessed violence within the family in their childhood reproduce this pattern [95,96]. This percentage includes children that in their adulthood are violent against women [97]. This could explain why physical abuse by family in childhood is a factor associated with IPF. Regarding stress theories, people have stress when they perceive an event as threatening and beyond their resources. On some occasions, the response to stressful situations is coped with by using violence, which can end in death [98,99]. Partner conflicts, unemployment, immigration, stepchildren, age difference, and low income have been identified as stressors that are associated with IPF [13,91].

Concerning crime opportunity theories, delinquency occurs at a time and place where there is a victim, a motivated aggressor, and an absence of control. Thus, crime is not randomly distributed [100]. This theory could explain why rural areas are associated with women’s deaths. In these areas, there is generally less availability of, accessibility to, and quality of professional services for victims [101,102,103]. The number of crime-control strategies is generally low and even absent in rural areas, and the same happens with the police [102,104].

It is important to point out that the factors identified in the review are not causes of IPF. The combination of factors from aggressor, victim, partner’s relationship, and environment could increase the probability of occurrence of fatal results according to the scientific studies. The factors are associated with the deaths, but they do not inevitably lead to said deaths. Thus, men cannot be criminalized, women cannot be blamed, and no individual attribute condemns the relationship or the environment. It is important to know the factors associated with the deaths to identify the cases with the highest risk of death and, consequently, to intervene and prevent the fatal result.

### 4.1. Limitations and Strengths

The systematic review has several limitations that must be contemplated in interpreting the results. Firstly, there is a low number of scientific studies on factors associated with IPF that refute and validate the current results. Thus, the results are not yet sufficiently consolidated. Secondly, three studies could not be obtained in full text and could meet the inclusion criteria. These could analyze more factors associated with IPF than those contemplated in this review. Thirdly, few studies used a prospective methodology, and most of them applied a retrospective methodology based on past data; it is difficult to follow the population over time to obtain solid results on the association of the factors found with IPF. In addition, retrospective studies are based on official data and victim and aggressor self-reports, including subjective non-corroborated data. Fourthly, several studies are confusing regarding the designation of risk factors and descriptive characteristics, denominating, on some occasions, risk factors as descriptive characteristics in cross-sectional studies. Longitudinal studies are needed to identify risk factors of IPF and to know more than the association of factors with these deaths, providing knowledge about the probabilities of the lethal results. Fifthly, studies are carried out in different countries, making it difficult to extrapolate the results equally to all parts of the world. Most studies are conducted in Western societies, with differentiated elements such as culture and law. Social contexts influence beliefs and behaviors about women, as well as legal responses to violence against women.

The systematic review has certain strengths. Firstly, it is one of the few studies that has paid attention exclusively to factors associated with women’s deaths by male intimate partners. This review focuses on a little-studied phenomenon, but it is necessary to understand it to improve prediction and prevention. Secondly, the current work provides updated findings that help obtain more comprehensive knowledge of these deaths. It synthesizes many factors associated with women’s deaths by the scientific literature, including the findings on factors from recent studies that have not been included in previous systematic reviews. Thirdly, the current research included quantitative, qualitative, and mixed studies. This allows the acquisition and integration of various factors of different kinds. Observable factors from official data and non-observable factors through self-reports provide valuable information concerning the factors associated with IPF. Fourthly, the PRISMA guideline was used, and it allowed carrying out a rigorous systematic process. Moreover, the MMAT guideline was also used for quality assessment with good results, contributing to a reliable validation of our findings.

### 4.2. Implications

The knowledge of factors associated with IPF facilitates the detection of women whose lives are in danger from different perspectives such as social, health, and judicial services. It helps them take early actions to prevent homicide. Preventive actions are also relevant before the presence of danger through programs for women and men that intervene based on dynamic factors associated with these deaths. This reduces the risk of being a victim and aggressor in the future. It is also important to prevent recidivism by focusing on aggressors and intervening based on their factors.

Furthermore, implementing criminal policies based on the findings is essential to determine effective law measures to prevent and react effectively to IPF. Adaptations to the type, time, and execution of sentences for crimes related to violence against women could be determinant to achieving the prevention and control intended in the law. Evidence is applied to social and health policies in institutions that support victims, enable resources, and empower them to end the maltreatment. Additionally, research into factors associated with IPF is needed, as they are not yet sufficiently refuted or validated. There is also a necessity for more research focused on unstudied factors, especially those related to the new pandemic context.

## 5. Conclusions

This systematic review makes a comprehensive contribution to the knowledge of factors associated with IPF, revealing that these encompass aggressor, victim, partner’s relationship, and environment. Therefore, these deaths relate to an extensive group of characteristics present in different people, situations, and places. They are not related to individual variables exclusively. The review provides updated information, validating previously known factors and identifying new ones contributing to predicting and preventing future IPF. More research is also needed, mostly on individual unstudied factors and clusters of factors that potentiate and mitigate the lethal result. The analysis of the contribution of factors to IPF in terms of probability is essential. For this purpose, more prospective cohort studies are required. In these future studies, cisgender and transgender victims of violence must be considered. The studies included described victims as females, but did not specify their sex and gender. This is important to know since transgender women are at a higher risk of maltreatment.

## Figures and Tables

**Figure 1 ijerph-19-07336-f001:**
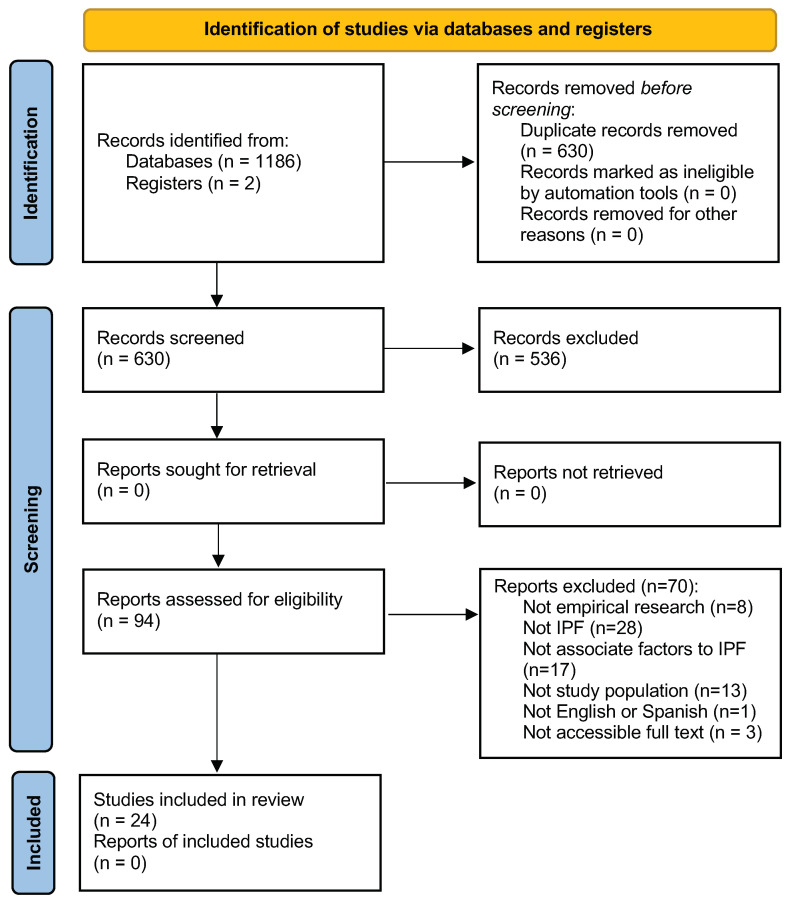
PRISMA flow diagram of the studies’ selection process [45].

**Table 2 ijerph-19-07336-t002:** MMAT checklist quality assessment [51].

Quantitative Descriptive Method Studies							
	Clear research questions	The collected data allow addressing the research questions	The sampling strategy is relevant to address the research questions	The sample is representative of the target population	The measurements are appropriate	The risk of nonresponse bias is low	The statistical analysis is appropriate to answer the research questions
[56]		X	X	X	X	X	X
[67]	X	X	X	X	X	X	X
[33]				X	X	X	X
[68]		X	X	X	X	X	X
[57]				X	X	X	X
[61]				X	X	X	X
[34]		X	X	X	X	X	X
[55]				X	X	X	X
[65]	X	X	X	X	X	X	X
[63]				X	X	X	X
[62]				X	X	X	X
[64]				X	X	X	X
[6]				X	X	X	X
[59]				X	X	X	X
[10]				X	X	X	X
[69]	X	X	X	X	X		X
[58]	X	X	X	X	X	X	X
[70]				X	X	X	X
**Qualitative Method Studies**							
	There are clear research questions	The collected data allow addressing the research questions	The qualitative approach is appropriate to answer the research questions	The qualitative data collection methods are adequate to address the research questions	The findings are adequately derived from the data	There is interpretation of results sufficiently substantiated by data	There is coherence between qualitative data sources, collection, analysis, and interpretation
[1]		X	X	X	X	X	X
[13]		X	X	X	X	X	X
[11]				X	X	X	X
[2]	X	X	X	X	X	X	X
**Mixed Method Studies**							
	There are clear research questions	The collected data allow addressing the research questions	There is an adequate rationale for using a mixed method design to address the research question	The different components of the study are effectively integrated to answer the research questions	The outputs of the integration of qualitative and quantitative components are adequately interpreted	Divergences and inconsistencies between quantitative and qualitative results are adequately addressed	The different components of the study adhere to the quality criteria of each of the methods involved
[60]	X	X	X	X	X	X	
[66]	X	X	X	X	X	X	X

## Data Availability

Not applicable.

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
