# Peer review of "A Comprehensive Analysis of Factors Associated with Intimate Partner Femicide: A Systematic Review"

_ijerph, 2022, doi:10.3390/ijerph19127336_

Round 1

Reviewer 1 Report

The present manuscript describes a systematic review of the literature on intimate partner femicide (IPF). The authors describe the state of the literature and discuss the myriad factors associated with IPF across countries. This work is relevant and sheds light on a relatively understudied phenomenon. My main critique is that the writing can be improved. To be clear, the writing is not poor. However, some of the sentences are unclear- I assume that English is not the first language of the authors. I would recommend publishing this manuscript after some revisions. See my specific comments below.

Methods

1.    It would be helpful to know if the authors are referring to cisgender women in their research. The authors only describe victims of IPF as being female and not in same sex relationships. This is relevant as research has shown transgender women to be at heightened risk for IPV.

2.    Line 98 – I am not clear on what is meant by “…(2) make offender’s classification of homicide by intimate 98partner violence…

3.    Table 1’s formatting makes it hard to read (e.g., spaces between words). Consider reformatting.

Discussion

1.    Line 341 the statement “…men with mental illness usually suffer from substance abuse (drug and alcohol) to cope with their difficulties…” is too broad of a generalization and not supported by the evidence.

2.    The authors correctly acknowledge that their findings are all associations and not causations. It may be helpful for the authors to discuss future research that could expand upon their findings. For example, through prospective cohort studies.

Author Response

A Comprehensive Analysis of Factors Associated with Intimate Partner Femicide: A Systematic Review

June 3, 2022

Esperanza Garcia-Vergara 1, Nerea Almeda 2, Blanca Martínn Ríos 3, David Becerra-Alonso 1 and Francisco Fern.ndez-Navarro 1

[email protected]

1Departament of Quantitative Methods, Universidad Loyola Andalucia, Seville, Spain

2Departament of Psychology, Universidad Loyola Andalucia, Seville, Spain

3Departament of Legal and Political Sciences, Universidad Loyola Andalucia, Seville, Spain

Acknowledgements

First of all, the authors would like to thank Reviewer for his/her helpful suggestions for this paper. All have been taken into account in revising and improving the manuscript. Below, the specific responses to the comments will be addressed.

Comments made by Reviewer 1

1 The present manuscript describes a systematic review of the literature on intimate partner femicide (IPF). The authors describe the state of the literature and discuss the myriad factors associated with IPF across countries. This work is relevant and sheds light on a relatively understudied phenomenon. My main critique is that the writing can be improved. To be clear, the writing is not poor. However, some of the sentences are unclear- I assume that English is not the first language of the authors. I would recommend publishing this manuscript after some revisions. See my specific comments below.

2 It would be helpful to know if the authors are referring to cisgender women in their research. The authors only describe victims of IPF as being female and not in same sex relationships. This is relevant as research has shown transgender women to be at heightened risk for IPV.

3 Line 98 – I am not clear on what is meant by “ (2) make offender′s classification of homicide by intimate partner violence.... ”

4 Table 1′s formatting makes it hard to read (e.g., spaces between words). Consider reformatting.

5 Line 341 the statement “ men with mental illness usually suffer from substance abuse (drug and alcohol) to cope with their difficulties” is too broad of a generalization and not supported by the evidence.

6 The authors correctly acknowledge that their findings are all associations and not causations. It may be helpful for the authors to discuss future research that could expand upon their findings. For example, through prospective cohort studies.

Responses to Reviewer 1

1 Thank you for appreciating our research work. The English have been thoroughly revised again.

2 Thank you for highlighting this. We are at this point unable to answer your comment. We have limited ourselves to the description of the studies included in the systematic review. These studies do not indicate whether they include only cisgender cases or whether they also involve transgender cases. They do not specify this aspect so we cannot know since our review is based on secondary sources (scientific studies). We agree on its importance, so we have included it for future research in the revised version of the manuscript (lines 483-486).

3 Thank you for pointing us to unclear information. We have changed the second exclusion criteria in the revised version of the manuscript (line 110).

4 The format of table 1 has been improved in the revised version of the manuscript.

5 This is an important remark. A more precise sentence has been included in the revised version of the manuscript (lines 388-390).

6 Thanks for the comment. We have discussed future research in the revised version of the manuscript (lines 481-483).

Reviewer 2 Report

Highly interesting topic and a review that has been compiled in a very professional and compact manner and will most likely serve the public as the topic treated is so concrete.

My comments concern mainly two things: first of all I would like a bit more precision why these particular data bases were chosen for the purpose of the review, was there some specific reason for that, and second of all, do you think that the fact that most of the studies are conducted in the Western societies should somehow be discussed while compiling the findings, is it of relevance when it comes to the results of the review?

Some more technical remarks.

rows 34-35 maybe too many verbs in this sentence

row 91 here would be interesting to know why these databases were chosen and how can they be accessed, if someone else wishes to use them

row 95 and 195 all criteria needed to be met?

row 139 “immigrant men can”, I do not think that one can state that “all immigrant men usually represent risk factors” without sounding biased

row 145 I think that here it would also be important to take a look at which kind of opportunities are offered to the migrants in the society, which kind of possibilities they have for social mobility and which kind of obstacles and frustrations they face

row 387 most of the studies are from Western societies, would be important to conduct studies elsewhere too

Author Response

A Comprehensive Analysis of Factors Associated with Intimate Partner Femicide: A Systematic Review

June 3, 2022

Esperanza Garcia-Vergara 1, Nerea Almeda 2, Blanca Martín Ríos 3, David Becerra-Alonso 1 and Francisco Fern.ndez-Navarro 1

[email protected]

1Departament of Quantitative Methods, Universidad Loyola Andalucia, Seville, Spain

2Departament of Psychology, Universidad Loyola Andalucia, Seville, Spain

3Departament of Legal and Political Sciences, Universidad Loyola Andalucia, Seville, Spain

Acknowledgements

First of all, the authors would like to thank Reviewer for his/her helpful suggestions for this paper. All have been taken into account in revising and improving the manuscript. Below, the specific responses to the comments will be addressed.

Comments made by Reviewer 2

1 Highly interesting topic and a review that has been compiled in a very professional and compact manner and will most likely serve the public as the topic treated is so concrete.

2 My comments concern mainly two things: first of all I would like a bit more precision why these particular data bases were chosen for the purpose of the review, was there some specific reason for that, and second of all, do you think that the fact that most of the studies are conducted in theWestern societies should somehow be discussed while compiling the findings, is it of relevance when it comes to the results of the review?

3 Rows 34-35 maybe too many verbs in this sentence.

4 Row 95 and 195 all criteria needed to be met?

5 Row 139 “immigrant men can”, I do not think that one can state that “all immigrant men usually represent risk factors” without sounding biased.

6 Row 145 I think that here it would also be important to take a look at which kind of opportunities are offered to the migrants in the society, which kind of possibilities they have for social mobility and which kind of obstacles and frustrations they face.

Responses to Reviewer 2

1 Thank you for appreciating our research work.

2 The databases used in the systematic review were chosen for their quality in the scientific field and theme according to the research topic. This has been specified in the revised version of the manuscript (lines 96-97). Regarding the Western society’s comment, there has been further discussion of the place of the studies as a limitation in the revised version of the manuscript (lines 437-441).

3 The English have been thoroughly revised again in the revised version of the manuscript (lines 34-35).

4 According to the PRISMA guideline, the studies included in the systematic review met all inclusion criteria. If any exclusion criteria are met, they are removed from the systematic review. This aspect is explained in detail in the revised version of the manuscript (lines 115-120). 

5 This is an important remark. Changes in the referred sentence have been made in the revised version of the manuscript (lines 188-191).

6 The aspects mentioned are relevant. Unfortunately, these cannot be included in the results section because the PRISMA guidelines only allow reporting information from the included studies, and these studies do not mention the opportunities, social mobility, obstacles, or frustrations of migrants. These aspects could be addressed in future studies.

Reviewer 3 Report

Dear authors, thank you for giving me the opportunity to review your manuscript: “A Comprehensive Analysis of Factors Associated with Intimate Partner Femicide: A Systematic Review”. I think that this manuscript can meaningfully contribute to the literature. 

I send below some comments.

Materials and Methods: I suggest changing to “methodology”

-       I suggest including “Search results” in the methodology section

-       I suggest changing “search results” to “Study selection.”

-       On “search results”, authors should give more details about excluding articles (for example "The title and abstracts of these were read and removed according to the inclusion and exclusion criteria.") Please, clarify why.

Results:

-        Authors should develop "Study characteristics" section, and don't just refer the table.

-        In the results section, I think it would be a plus to add the characteristics of the study sample (age, marital status, etc.).

Discussion

-        In the limitations section, the authors should add that they cannot access 3 articles in full.

-        The authors state that: “Fourthly, most studies are carried out in different countries, making it difficult to extrapolate the results equally to all parts of the world.” The authors should elaborate on this idea, explaining how this can be a limitation. What are the specifics of each country that might make this factor a limitation for the study?

Author Response

A Comprehensive Analysis of Factors Associated with Intimate Partner Femicide: A Systematic Review

June 3, 2022

Esperanza Garcia-Vergara 1, Nerea Almeda 2, Blanca Martín Ríos 3, David Becerra-Alonso 1 and Francisco Fernández-Navarro 1

[email protected]

1Departament of Quantitative Methods, Universidad Loyola Andalucia, Seville, Spain

2Departament of Psychology, Universidad Loyola Andalucia, Seville, Spain

3Departament of Legal and Political Sciences, Universidad Loyola Andalucia, Seville, Spain

Acknowledgements

First of all, the authors would like to thank Reviewer for his/her helpful suggestions for this paper. All have been taken into account in revising and improving the manuscript. Below, the specific responses to the comments will be addressed.

Comments made by Reviewer 3

1 Dear authors, thank you for giving me the opportunity to review your manuscript: “A Comprehensive Analysis of Factors Associated with Intimate Partner Femicide: A Systematic Review”. I think that this manuscript can meaningfully contribute to the literature.

2 Materials and Methods: I suggest changing to “methodology”.

3 I suggest including “Search results” in the methodology section. I suggest changing “search results” to “Study selection”.

4 On “search results”, authors should give more details about excluding articles (for example “The title and abstracts of these were read and removed according to the inclusion and exclusion criteria”) Please, clarify why.

5 Authors should develop “Study characteristics” section, and don’t just refer the table.

6 In the limitations section, the authors should add that they cannot access 3 articles in full.

7 The authors state that: “Fourthly, most studies are carried out in different countries, making it difficult to extrapolate the results equally to all parts of the world”. The authors should elaborate on this idea, explaining how this can be a limitation. What are the specifics of each country that might make this factor a limitation for the study?

Responses to Reviewer 3

1 Thank you for appreciating our research work.

2 The proposed denomination is changed in the revised version of the manuscript (lines 83).

3 According to the PRISMA guidelines, the section “search results” is the correct denomination for systematic reviews and meta-analysis. Furthermore, the same section cannot be included in the methodology because it is part of the results following the mentioned guideline.

4 The reasons for excluding the studies are explained in the method section (lines 110-114). All this is done under the PRISMA guidelines.

5 Thanks for the suggestion. A “Study characteristics” section has been included in the revised version of the manuscript (lines 141-174).

6 Thanks for the comment. The limitation of not being able to access three articles in full has been included in the revised version of the manuscript (lines 426-428).

7 We have clarified the referred limitation in the revised version of the manuscript (lines 437-441).

Reviewer 4 Report

Comments to the Author

This manuscript, titled “A Comprehensive Analysis of Factors Associated with Intimate Partner Femicide: A Systematic Review,” used a systematic review approach to analyze aggressor-related, victim-related, partner relationship-related, and environmental-related factors in intimate partner femicide (IPF). The review paper investigated an important research question that carries significant practical implications from a comprehensive perspective.

There are several issues that need to be addressed.

  • The Introduction could be further enriched by giving a more comprehensive review of the literature. As the aim of the systematic review was to focus on the aggressor-, victim-, partner relationship-, and environmental-related factors on IPF, a review is necessary to discuss how these factors are relevant to IPF. While it is good to highlight the gaps in Contreras (2014) and Spencer and Stith (2020), it would be helpful to report the conclusions/findings in their papers.
  • It is not clear the range of publication years used in the literature search and who performed the search.
  • Exclusion criteria #1-#3 are hard to follow. As such, it is not clear why Reference #12 was not included in the reviewed paper list and subsequently why “attachment” was not reported in the Results section.
  • The results section could benefit from more clarity, for instance:
    • Line 149: “this type of death” – It is not clear what “this type” refers to.
    • Lines 159-160: Hard to follow.
    • Lines 231-232: “This increases the probabilities…” – What does “this” refer to?
  • In the Results section, it was not clear which studies cited in the text were the papers included for the systematic review (unless checked against Table 2) and which papers were cited just to provide support for arguments. Therefore, the authors are recommended to be explicit about it. Section 3.2.3 was a good example.
    • For Section 3.2.1, it could be amended to something along the line of: “Several studies [1, 33, 49] showed that perpetrators’ SES was associated with the likelihood of IPF. Low-medium socio-economic status is a factor associated with perpetrating this crime [49]……”
  • Line 200: “The quality is different across the quantitative, qualitative, and mixed methods studies” – It would be helpful to be explicit about which types of studies had higher quality. It is not clear which author(s) or personnel performed the quality assessment. If more than 2 persons were involved, how were discrepancies resolved?
  • The discussion section was well-written with a theory-based discussion of theoretical implications, limitations, and implications.
  • The papers that met the inclusion criteria for the systematic review should be labeled with an asterisk in the reference list.
  • Just a suggestion: The lack of homogeneity (line 80) could mean there are moderators in the association between the factors of interest and IPF. The authors could explore further using meta-analyses by including moderators in a future study.

Author Response

A Comprehensive Analysis of Factors Associated with Intimate Partner Femicide: A Systematic Review

June 3, 2022

Esperanza Garcia-Vergara 1, Nerea Almeda 2, Blanca Martín Ríos 3, David Becerra-Alonso 1 and Francisco Fern.ndez-Navarro 1

[email protected]

1Departament of Quantitative Methods, Universidad Loyola Andalucia, Seville, Spain

2Departament of Psychology, Universidad Loyola Andalucia, Seville, Spain

3Departament of Legal and Political Sciences, Universidad Loyola Andalucia, Seville, Spain

Acknowledgements

First of all, the authors would like to thank Reviewer for his/her helpful suggestions for this paper. All have been taken into account in revising and improving the manuscript. Below, the specific responses to the comments will be addressed.

Comments made by Reviewer 4

1 This manuscript, titled “A Comprehensive Analysis of Factors Associated with Intimate Partner Femicide: A Systematic Review”, used a systematic review approach to analyze aggressor-related, victim-related, partner relationship-related, and environmentalrelated factors in intimate partner femicide (IPF). The review paper investigated an important research question that carries significant practical implications from a comprehensive perspective.

2 The Introduction could be further enriched by giving a more comprehensive review of the literature. As the aim of the systematic review was to focus on the aggressor-, victim-, partner relationship-, and environmental-related factors on IPF, a review is necessary to discuss how these factors are relevant to IPF. While it is good to highlight the gaps in Contreras (2014) and Spencer and Stith (2020), it would be helpful to report the conclusions/findings in their papers.

3 It is not clear the range of publication years used in the literature search and who performed the search.

4 Exclusion criteria 1-3 are hard to follow. As such, it is not clear why Reference 12 was not included in the reviewed paper list and subsequently why “attachment” was not reported in the Results section.

5 The results section could benefit from more clarity, for instance: Line 149: “this type of death” It is not clear what “this type” refers to. Lines 159-160: Hard to follow. Lines 231-232: “This increases the probabilities” What does “this” refer to?

6 In the Results section, it was not clear which studies cited in the text were the papers included for the systematic review (unless checked against Table 2) and which papers were cited just to provide support for arguments. Therefore, the authors are recommended to be explicit about it. Section 3.2.3 was a good example. For Section 3.2.1, it could be amended to something along the line of: “Several studies [1, 33, 49] showed that perpetrators SES was associated with the likelihood of IPF. Low-medium socioeconomic status is a factor associated with perpetrating this crime [49]”

7 Line 200: “The quality is different across the quantitative, qualitative, and mixed methods studies” It would be helpful to be explicit about which types of studies had higher quality. It is not clear which author(s) or personnel performed the quality assessment. If more than 2 persons were involved, how were discrepancies resolved?

8 The papers that met the inclusion criteria for the systematic review should be labeled with an asterisk in the reference list.

9 Just a suggestion: The lack of homogeneity (line 80) could mean there are moderators in the association between the factors of interest and IPF. The authors could explore further using meta-analyses by including moderators in a future study.

Responses to Reviewer 4

1 Thank you for appreciating our research work.

2 The proposed modifications are changed in the revised version of the manuscript (lines 54-57 and 67-76).

3 Thanks for the valuable comment. The search strategy did not apply limitations to the range of publication years of the studies, specifying them in the revised version of the manuscript (lines 94 and 104).

4 Thank you for pointing us to unclear information. We have rewritten the exclusion criteria number 1 and 3 in the revised version of the manuscript (lines 110-113). Additionally, the study with reference number 12 has not been included in the systematic review because it refers to the typology of intimate partner homicide and is an exclusion criterion.

5 We have revised and rewritten the referred parts of the results section in the revised version of the manuscript (lines 197-198, lines 206-208, lines 279-280).

6 According to the PRISMA guidelines, the studies cited in the results section are included in the systematic review. The tables explicitly list the studies included.

7 It is not possible to report which types of studies have higher quality. The quality assessment of the included studies allows the analysis of the risk of bias, not judging how good or bad they are. Additionally, we have clarified the researchers’ discrepancies in the revised version of the manuscript (lines 131-132).

8 Thank you for the suggestion. We have included an asterisk in the reference list.

9 Thanks for the valuable comment. We agree with the suggestion, and we have incorporated it for future research in the revised version of the manuscript (lines 481-482).

Round 2

Reviewer 4 Report

The authors are commended for their effort put forth in revising the manuscript.

Author Response

Dear Editor, the authors would like to thank you for your excellent work supporting our research. Following the reviewer's recommendations, the manuscript has been reviewed by two native English speakers. The corrections have been included in red in the revised version of the manuscript.  If specific English changes are required, please provide us with the pages and lines numbers please.